# The influence of neuromuscular blockade on phase lag entropy and bispectral index: A randomized, controlled trial

Sohee Jin[1], Hee Jung Baik[2]*, Sooyoung Cho[2], Rack Kyung Chung[2], Kyoung Ae Kong[3], Youn Jin Kim[4]

1 Department of Anesthesiology and Pain Medicine, Yonsei Barun Hospital, Seoul, Republic of Korea, 2 Department of Anesthesiology and Pain Medicine, College of Medicine, Ewha Womans University, Ewha Womans University Mokdong Hospital, Seoul, Republic of Korea, 3 Department of Preventive Medicine, College of Medicine, Ewha Womans University, Seoul, Republic of Korea, 4 Department of Anesthesiology and Pain Medicine, College of Medicine, Ewha Womans University, Ewha Womans University Seoul Hospital, Seoul, Republic of Korea

* baikhj@ewha.ac.kr

## Abstract

The aim of this study is to compare the effects of neuromuscular blockade (NMB) on phase lag entropy (PLE) and the bispectral index (BIS). We recorded the BIS, electromyograph (EMG) activity on a BIS monitor (EMG_BIS), PLE, and EMG activity on a PLE monitor (EMG_PLE) in 40 patients receiving general anesthesia. During the awake state, we analyzed the changes in parameters before and 2 min after the eyes were closed. During sedation, we compared the changes in the parameters before and at 4 min after injecting rocuronium (group R) or normal saline (group C) between the two groups. During anesthesia, we compared the changes in parameters before and at 4 min after injecting sugammadex (group B) or normal saline (group D) between the two groups. During the awake state, the BIS, EMG_BIS, and EMG_PLE, but not PLE, decreased significantly with closed eyes. An effect of EMG on the BIS was evident, but not on PLE. During sedation, the BIS decreased with the decrease in EMG_BIS regardless of NMB caused by rocuronium, but NMB decreased PLE, although the degree of the decrease in EMG_PLE after NMB was similar to that after placebo. To determine the effect of NMB on electroencephalograms (EEGs) in groups R and C, we plotted the power spectra before and at 4 min after injecting rocuronium or normal saline. Changes in slow and delta frequency bands were observed at 4 min after injecting rocuronium relative to before injecting rocuronium. There was no effect of EMG on either the BIS or PLE during anesthesia. In conclusion, the effect of electromyograph activity and/or neuromuscular blockade on BIS or PLE depends on the level of consciousness.

## Introduction

Various monitoring devices have been used to measure the depth of sedation and anesthesia in clinical practice. The bispectral index (BIS) is a processed electroencephalogram (EEG)

**Data Availability Statement:** All relevant data are within the manuscript and its Supporting Information files.

**Funding:** HJB: received grant from InBody Corporation, and the Industrial Strategic Technology Development Program (N10047988, 2013) funded by the Ministry of Industry and Trade. A neuroscience researcher (Kyoung-Soo Kim, from InBody)performed the spectral and Phase lag entropy data analyses. The funders had no role in study design, decision to publish, preparation of the manuscript, data collection and analysis, except spectral and phase lag entropy data analysis.

**Competing interests:** I have read the journal's policy and the authors of this manuscript have the following competing interests: This work was supported by InBody Corporation, and the Industrial Strategic Technology Development Program (N10047988, 2013) funded by the Ministry of Industry and Trade. This research was made possible by support from InBody who gifted the PLEM100 device and PLEM-ES100 electrode. This does not alter our adherence to PLOS ONE policies on sharing data and materials.

parameter and a quantifiable measure of the depth of sedation and anesthesia. It is a single score on a scale from 0 to 100 that is calculated from three subparameters: the Burst suppression ratio (BSR), the BetaRatio, and SynchFastSlow [1, 2]. In the BIS calculation, the weights of BSR and SynchFastSlow are greater during general anesthesia and the weight of BetaRatio is greatest under light sedation [3]. The BetaRatio is calculated as the log of the power ratio in two frequency bands {log (P30–40 Hz/P11–20 Hz)}, which include the frequencies generated by muscle activity. That is why neuromuscular blockade (NMB) affects the BIS [3]. Although neuromuscular blocking agents (NMBAs) affect the BIS, NMB reduces the BIS during sedation or light general anesthesia [3] but does not affect it during general anesthesia [4]. Therefore, it is necessary to consider the effects of NMB when reading BIS levels during sedation. NMBAs change the BIS in two ways. One is the interference effect on the BIS resulting from electromyograph (EMG) activity, and the other is that the muscle stretch receptor stimulates the arousal center of the brain (afferentation theory) [5]. However, this argument is controversial.

A device (PLEM™) has been developed that calculates phase lag entropy (PLE), a measure of the diversity of temporal patterns in the phase relationship between two signals. The PLEM is different from conventional devices, which do not provide information on communication between brain regions after analyzing single-channel EEG signals. The PLEM predicts the complexity of the communication between brain regions by calculating information entropy after extracting the phase relationship pattern; this can then be used to evaluate the depth of sedation and anesthesia. A previous study calculated PLE between two EEG signals from frontal and prefrontal montages (left: AF3-Fp1, right: AF4-Fp2); direct current offset was determined by subtracting the average amplitude of every 4-s epoch [6]. In another study, eye blink and high-amplitude (>100 µV) artefacts were removed from EEG signals [7].

PLE is composed of three sub-parameters: PLE1, PLE2, and BSR. PLE1 and PLE2 are calculated in alpha (8–13 Hz) and beta (13–30 Hz) bands for 4-s epochs without overlap, and slow frequency (0.1–1 Hz) and gamma (30–45 Hz) bands for 8-s epochs with a 50% overlap, respectively. PLE1 reflects a light hypnotic state, while PLE2 reflects a surgical hypnotic state. BSR includes two types of burst-suppression detection: the portions of the isoelectric EEG and the very low power frequency for 60 s [6, 8, 9]. PLE is calculated by combining PLE1, PLE2, and BSR with appropriate weights, which are linearly scaled from 0 to 100 [6, 9]. Although the frequency ranges of PLE2 are 0.1–1 Hz and 30–45 Hz, which include an EMG frequency band of $\geq$ 30 Hz, PLE does not use power spectrum analyses, unlike the BIS, which adopts the BetaRatio.

Therefore, we hypothesized that NMB affects BIS, but not PLE, during sedation state. And we also hypothesized that NMB does not affect BIS and PLE during a general anesthetic state.

The primary objective of this study was to evaluate the effects of NMB on PLE and BIS according to sedation status. The changes in the BIS and PLE after administering rocuronium, an NMBA, in the sedation state (BIS 60–80) during induction of anesthesia were compared to those of the placebo group, and the changes after administering sugammadex, an NMB reversal agent, during a general anesthetic state (BIS 40–55) were also evaluated. The secondary objectives of this study were to evaluate the effect of closing eyes on the BIS, PLE, EMG on BIS monitor (EMG_BIS), and EMG on PLE monitor (EMG_PLE) in the awake state and the effect of NMB on the EMG_BIS and EMG_PLE during sedation state and general anesthetic state.

## Materials and methods

This study was conducted at Ewha Womans University Mokdong Hospital from July 14th to September 21st, 2017. Forty patients aged 19–60 years, ASA physical status 1 or 2, scheduled for various elective surgeries under general anesthesia were enrolled (by S. Jin). The

institutional review board of Ewha Womans University Mokdong Hospital (IRB No. 2017–05–072–002) approved the study protocol on July 3rd, 2017 and written informed consent was obtained from all patients. This trial was registered at the Clinical Trial Registry of Korea (http://cris.nih.go.kr, KCT0003750) after enrollment of participants started due to delay in the preparation of document in English. Patients were excluded if they had cardiopulmonary, hepatic, renal, neurological, or neuromuscular disorders; if they had a history of any central nervous system disease, chronic alcohol consumption or drug abuse; or if they were taking any medications that affected neurological or neuromuscular function. Surgical procedures associated with anticipated major blood loss or fluid shift were also excluded.

## Randomization

The patients were randomized into group R (rocuronium 0.6 mg/kg injected intravenously) or C (same volume of saline) during induction of anesthesia. Separately, they were also randomized into group B (sugammadex 2 mg/kg injected intravenously) or D (same volume of saline) at the end of the operation. These two randomization processes were independent each other. The experimental group and the control group were randomly assigned using a computer-generated random number table (www.randomization.com) with block size 8 created by a nurse who prepared the study drugs according to the groups allocated. The patients, anesthesiologists involved in the study, who collected the data, and who performed statistical analysis were blinded to the group allocations.

## Interventions

After arrival in the operating room without premedication, noninvasive blood pressure, the electrocardiogram, and oxygen saturation ($SpO_2$) (using a pulse oximeter) were monitored. NMB was assessed by train-of-four (TOF) stimulation of the ulnar nerve using a Philips IntelliVue neuromuscular transmission (NMT) module (Philips, Best, The Netherlands), two skin electrodes (Red Dot®, 3M Health Care, Neuss, Germany), and an NMT acceleration sensor attached to the volar side of the ipsilateral thumb. After loss of consciousness, we calibrated the instruments, and then the NMT module automatically searched for the current needed for a supramaximal stimulus and maintained that current throughout the procedure. TOF monitoring was performed automatically, before and every minute after injecting a study drug during induction and at the end of the surgery, as well as every 10 min during maintenance of anesthesia.

A BIS sensor (BIS Quatro™ Sensor, Covidien, Mansfield, MA, USA) and a PLE sensor (special composite electrode with five elements) were attached to the patient's forehead. The BIS and PLE were measured using a BIS™ Vista A-3000 monitor (Aspect Medical Systems, Inc., Newton, MA, USA; software version 3.20) and a PLEM™ monitor (PLEM100, InBody Co., Seoul, Korea), respectively. S1 Fig shows the PLE and BIS sensors attached to a patient's forehead. After a 3–5 min stabilization period to achieve a signal quality index (SQI) of the BIS and PLE > 80, we recorded the baseline BIS, EMG_BIS, PLE, and EMG_PLE in the eye-open state (Tbase). Then we measured the same parameters at 2 min after the patient's eyes closed (Teye-closed).

The pharmacokinetic models of Schnider and Minto were used for propofol and remifentanil target-controlled infusion (TCI), respectively. After administering 0.2 mg glycopyrrolate, remifentanil TCI was started at a 2 ng/ml target effect-site concentration (Ce). After a stable remifentanil Ce was achieved, propofol TCI was started at a target Ce of 2 μg/ml, which was increased by 0.5 μg/ml until loss of consciousness (LOC). LOC was defined as no response to verbal commands to open the eyes, and was assessed every 15 s. When LOC was attained (T0),

the propofol Ce was held constant during the study period at induction and the BIS, EMG_-BIS, PLE, EMG_PLE, TOF count and ratio, blood pressure, and heart rate were recorded at 1-min intervals for 3 min (T1, T2 and T3). Then, 0.6 mg/kg rocuronium (group R) or the same volume of saline (group C) was injected intravenously. To maintain end-tidal partial pressure of $CO_2$ at 35–40 mmHg, assisted manual ventilation was performed with 100% oxygen via a face mask, and all measurements were recorded before and after administration of rocuronium at 1-min intervals for 4 min (TR0, TR1, TR2, TR3 and TR4). Thereafter, the target Ce of remifentanil was increased to 4 ng/ml and the target Ce of propofol was increased to maintain a BIS of 40–55. In group C and R, 0.6 mg/kg rocuronium and the same volume of normal saline were administered, respectively; 1.5 min later intubation was performed and anesthesia was maintained.

At the end of the operation, while the constant Ce of propofol was maintained to keep the BIS at 40–55 with a Ce of remifentanil of 2 ng/ml and TOF count $\geq$ 3 or a TOF ratio $<$ 20%, 2 mg/kg (group B) sugammadex (Bridion®, MSD, Seoul, Korea) or the same volume of saline (group D) was administered when three or more counts appeared on the TOF. We recorded all of the same measurements as induction before and after drug administration for 4 min at 1-min intervals (TRV0, TRV1, TRV2, TRV3, and TRV4) and thereafter, patients in groups D and B were given 0.4 mg glycopyrrolate and 10 mg pyridostigmine or the same volume of normal saline, respectively, and were awakened from anesthesia by stopping the TCI.

## Outcomes

The primary outcomes were variables of BIS, PLE, differences of BIS and PLE between TR0 and TR4 {$\Delta$(TR0 –TR4)}, and differences of BIS and PLE between TRV0 and TRV4 {$\Delta$(TRV0 –TRV4)}. The secondary outcomes included EMG_BIS, EMG_PLE, PLE1, PLE2, hemodynamic parameters, and the power of each EEG waves.

The BIS and PLE data were saved directly from the monitors to a USB flash drive, downloaded to a computer, and opened as an Excel file to extract the necessary data and confirm the handwritten collected data. The subject identification information in the saved data to USB flash drive and the handwritten data was encoded to protect personal information of participants in this study. PLE1 and PLE2 were calculated from the data using a MATLAB computer program (2017b, Mathworks Inc., Natick, MA, USA) with assistance from the PLEM™ manufacturer.

We performed spectral analyses of the EEGs in the PLEM™ if needed. The spectral analyses were performed using frequency information among the signal characteristics. The spectrum quantifies the energy in the signal or the power distribution by each frequency. Spectral analyses were calculated using MATLAB (version 2017.b, Mathworks Inc.). We used standard multi-taper power spectral density [10] estimated using the MATLAB signal-processing toolbox. We used a PLE sensor with a sampling rate of 128 Hz and a preamplifier bandwidth of 0.5–45 Hz to record the raw EEG signals. Electrode impedance in all channels was <7 kΩ. We collected EEG data for T0–T3 and TR0–TR4 in all patients for the spectral analyses during induction. For the power spectrogram, we obtained individual four-channel EEG signals using a PLE monitor (PLEM100, Inbody Co., Ltd., Seoul, Republic of Korea). The power spectrogram quantifies the frequency distribution of energy or power within the EEG signal over time. We computed group-level spectrograms for each group by taking the median across all time epochs and patients. We also computed group-level spectral curves by averaging across all time epochs and patients. The spectrogram parameters were window length = 2 s with overlap = 15 s (75%), time resolution of 0.5 s, and spectral resolution of 0.25 Hz.

### Statistical analyses

**Sample size.** The sample size was calculated based on Cohen's formula with two-sided testing using G\*Power program version 3.1.9.2. Applying the effect size (1.04) calculated by the change of BIS after NMB (relaxant group: 12.9 ± 6.2, placebo group: 6.2 ± 6.7) of a previous study [11], the number of patients needed to provide a significance level (α error) of 0.05 and a statistical power (1 –β error) of 0.8 was 32. Forty patients were enrolled to allow for a dropout rate of up to 20%.

**Statistical methods.** The normality of continuous data was tested with Kolmogorov-Smirnov test and Q-Q plot evaluation. Data are expressed as mean ± SD, median (interquartile range), number, or percentage of patients, as appropriate. Comparisons of data before and after the eyes closed were analyzed using paired Student's t-tests. Intragroup continuous data across time were analyzed using repeated-measures analysis of variance (ANOVA), with a post hoc Bonferroni correction for multiple comparisons. Intragroup comparisons of data between TR0 and TR4 or between TRV0 and TRV4 were performed using the Wilcoxon rank-sum test. Intergroup comparisons of data at TR0, TR4, the data differences between TR0 and TR4 {Δ (TR0 –TR4)}, data at TRV0, at TRV4, and the data differences in TRV0 and TRV4 {Δ(TRV0 – TRV4)} were analyzed using the Mann–Whitney U-test. P-values < 0.05 were considered significant.

## Results

The enrolment flow chart shows the number of patients enrolled and analyzed during the three study periods (Fig 1).

### Study during induction of anesthesia

Fig 2A shows the time flow for the experimental procedure during induction of anesthesia.

**The effect of closing eyes on BIS and PLE during the awake state.** Table 1 shows the changes in the BIS, EMG_BIS, PLE, EMG_PLE, PLE1, and PLE2 before and after the eyes closed. The BIS, EMG_BIS, and EMG_PLE decreased significantly after the eyes closed, but PLE, PLE1, and PLE2 did not change. Mean difference (95% confidence interval (CI)) of BIS

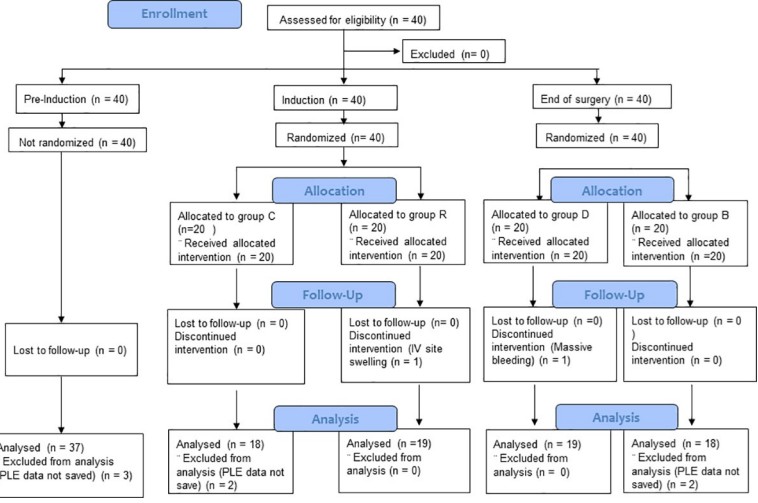

**Fig 1. CONSORT flow diagram.**

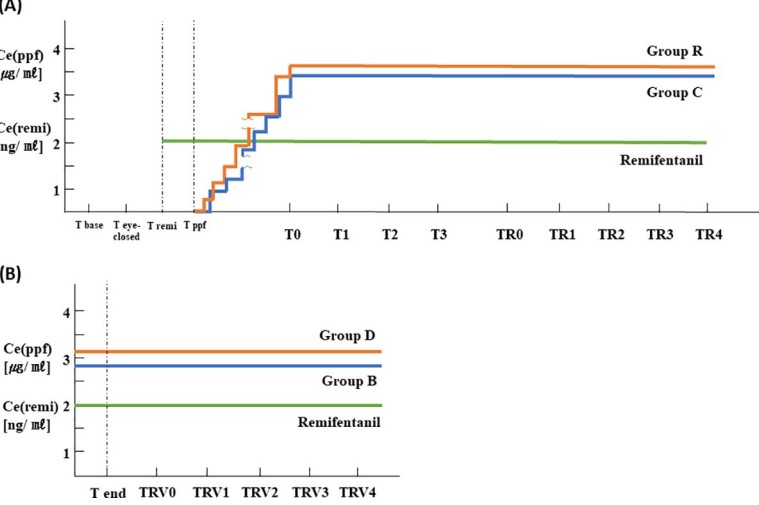

**Fig 2. Time flow of the study at the induction and the end of surgery.** (A) Ce (ppf): effect-site concentration of propofol, Ce (remi): effect-site concentration of remifentanil, Tbase: at baseline (in the eye-open state); Teye-closed: at 2 min after the patient's eyes closed; Tremi: time when remifentanil TCI started; Tppf: time when propofol TCI started, T0: time when loss of consciousness (LOC) occurred; T1, T2 and T3: 1, 2 and 3 min after LOC, respectively, TR0, TR1, TR2, TR3 and TR4: immediately before, and at 1, 2, 3 and 4 min after injecting rocuronium (group R) or the same volume of normal saline (group C), respectively. (B) Ce (ppf): effect-site concentration of propofol, Ce (remi): effect site concentration of remifentanil, Tend: at the end of surgery, TRV0, TRV1, TRV2, TRV3 and TRV4: immediately before and at 1, 2, 3 and 4 min after injecting sugammadex (group B) or the same volume of normal saline (group D), respectively.

and EMG_BIS, PLE, and PLE_EMG between $T_{base}$ and $T_{eye-closed}$ were 5.7 (3.94–7.46) and 1.85 (0.66–3.04), 1.27 (-0.22–2.76), and 11.08 (8.71–13.46), respectively.

**The changes of BIS and PLE for 3 minutes while propofol Ce at LOC was maintained.** The demographic data and clinical characteristics during induction of anesthesia are shown in Table 2. During induction of anesthesia, propofol Ce at LOC was 3.5 ± 0.6 μg/ml in 37 patients. The BIS, EMG_BIS, PLE, EMG_PLE, PLE1, and PLE2 decreased significantly at all time points compared to T0 ($P < 0.05$) while propofol Ce at LOC was maintained for 3 min. The BIS, EMG_BIS, PLE, EMG_PLE and PLE1, but not PLE2, decreased significantly for 3 min ($P < 0.05$) (Table 3). Mean blood pressure decreased significantly at all time points compared to T0 (T0: 82.1 ± 10.9 mmHg, T1: 74.9 ± 12.4 mmHg, T2: 73.4 ± 11.3 mmHg, T3: 72.5 ± 13.7 mmHg, $P < 0.05$). Heart rate decreased significantly only at T3 compared to T0 (T0: 69.2 ± 10.4 /min, T1: 67.8 ± 9.9/ min, T2: 67.2 ± 10.0 /min, T3: 66.0 ± 9.3 /min, $P < 0.05$). We

**Table 1. Changes before and after closing eyes.**

|  ' | $T_{base}$ (n = 37) | $T_{eye-closed}$ (n = 37) |
|---|---|---|
| BIS | 95.2 ± 2.7 | 89.5 ± 5.2 * |
| EMG_BIS | 48.1 ± 4.3 | 46.2 ± 4.9 * |
| PLE | 87.7 ± 3.9 | 86.4 ± 3.5 |
| EMG_PLE | 46.0 ± 11.3 | 34.9 ± 10.6 * |
| PLE1 | 0.88 ± 0.04 | 0.86 ± 0.05 |
| PLE2 | 0.78 ± 0.16 | 0.80 ± 0.11 |

Values are mean ± SD. BIS; Bispectral index, PLE; Phase lag entropy, EMG; Electromyography, $T_{base}$ and $T_{eye-closed}$: baseline and 2 min after closing eyes, respectively.

*: $P < 0.05$, compared with $T_{base}$

**Table 2. Demographic and clinical data in the study during induction.**

| | Group C | Group R |
|---|---|---|
| | (n = 18) | (n = 19) |
| Age (years) | 46.2 ± 12.7 | 45.8 ± 11.1 |
| Gender (Male: Female) | 12: 6 | 11: 8 |
| Weight (kg) | 66.9 ± 11.1 | 63.2 ± 10.0 |
| Height (cm) | 167.3 ± 8.4 | 166.5 ± 9.0 |
| Surgical department {N (%)) | | |
| OS | 10 (55.6) | 8 (42.1) |
| URO | 4 (22.2) | 5 (26.3) |
| GS | 3 (16.7) | 5 (26.3) |
| OBGY | 1 (5.6) | 1 (5.3) |

Values are mean ± SD or number (% of group). Group C: group who received same volume of normal saline as rocuronium, Group R: group who received rocuronium 0.6 mg/kg. There were no significant differences between the two groups.

performed spectral analyses of saved EEGs in PLEM™ data to determine whether the gradual decreases in the BIS and PLE for 3 min while maintaining a constant Ce of propofol were related to changes in EEG waves reflecting the depth of sedation/anesthesia. The group-median spectrogram showed that the power of the alpha and beta waves became stronger and weaker, respectively, over time (Fig 3).

**The changes of BIS and PLE for 4 minutes after administration of rocuronium.** No significant differences were observed in propofol Ce at LOC, which was constantly maintained in all patients during the study period between group C (3.4 ± 0.8 μg/ml) and R (3.6 ± 0.6 μg/ml) (P > 0.05) (Fig 2A). The BIS immediately before administering rocuronium or saline (at TR0) to group C or R was 59.3 ± 9.9 or 60.0 ± 5.6, respectively, however, no significant differences were observed between the two groups (P > 0.05) (Table 4). At 2, 3, and 4 min after injecting saline in group C and 1, 2, 3, and 4 min (at TR1, TR2, TR3, and TR4) after injecting rocuronium in group R, the BIS decreased significantly compared to TR0 (P < 0.05), but no significant differences were observed between the two groups at any of the time points (P > 0.05). The EMG_BIS at TR1, TR2, TR3, and TR4 decreased significantly compared to TR0 only in group R (P < 0.05). The EMG_BIS in group R was significantly lower than that in group C at TR2 and TR3 (P < 0.05) (Table 4). PLE values at TR0 in group C and group R were 53.4 ± 8.4

**Table 3. Changes in BIS and PLE for 3 min while propofol Ce at LOC Was maintained before neuromuscular blockade.**

| | T0 (n = 37) | T1 (n = 37) | T2 (n = 37) | T3 (n = 37) |
|---|---|---|---|---|
| BIS | 75.9 ± 8.1 | 69.4 ± 9.1 *† | 66.3 ± 7.0 *† | 61.6 ± 6.6 *† |
| EMG_BIS | 45.1 ± 7.0 | 40.3 ± 7.9 *† | 36.4 ± 5.9 *† | 34.0 ± 5.1 *† |
| PLE | 70.1 ± 8.6 | 62.7 ± 11.6 *† | 59.6 ± 8.7 * | 56.8 ± 8.3 *† |
| EMG_PLE | 33.7 ± 14.1 | 27.2 ± 10.5 *† | 23.4 ± 6.9 *† | 22.5 ± 6.9 * |
| PLE1 | 0.69 ± 0.93 | 0.65 ± 0.67 *† | 0.61 ± 0.05 *† | 0.60 ± 0.04 * |
| PLE2 | 0.73 ± 0.17 | 0.66 ± 0.16 * | 0.62 ± 0.13 * | 0.60 ± 0.12 * |

Values are mean ± SD. BIS; Bispectral index, PLE; Phase lag entropy, EMG; Electromyography, T0: the time when loss of consciousness (LOC) occurred, T1, T2, and T3: 1, 2, and 3 min after LOC, respectively.

*: P < 0.05 compared with T0.

†: P < 0.05 compared with previous value.

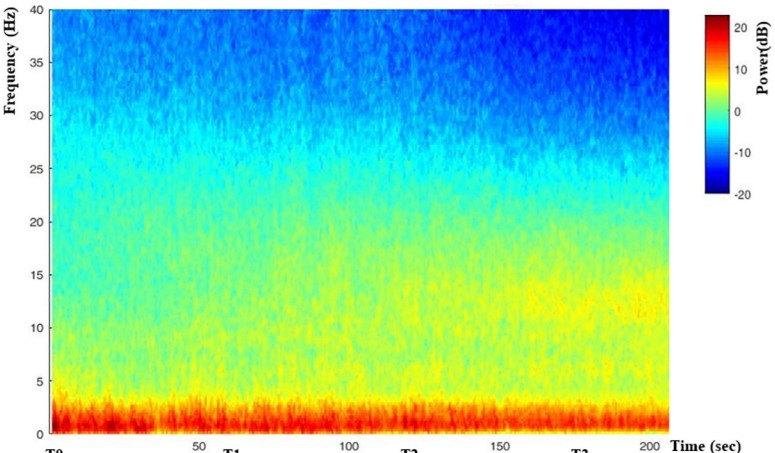

**Fig 3. Group-median spectrogram during T0–T3.** Group-median spectrograms show that the power of alpha waves (8–13 Hz) and beta waves (13–30 Hz) became stronger and weaker, respectively, with time.

and 57.1 ± 8.1, respectively, however, no significant differences were observed between the two groups (P > 0.05). PLE decreased significantly compared to TR0 in group R (P < 0.05) at TR2, TR3, and TR4. However, no significant differences were observed in PLE or EMG_PLE between the two groups at any time point (P > 0.05) (Table 4). Mean blood pressure and heart rate did not change significantly at any time point compared to TR0 in either group, and no significant differences were observed between the two groups (S1 Table).

**Table 4. Changes in BIS and PLE after administration of study drug.**

|  | Group | TR0 | TR1 | TR2 | TR3 | TR4 |
|---|---|---|---|---|---|---|
| BIS | C (n = 18) | 59.3 ± 9.0 | 55.6 ± 9.9 | 52.4 ± 7.4 † | 50.7 ± 7.9 † | 47.1 ± 8.0 †‡ |
|  | R (n = 19) | 60.0 ± 5.6 | 55.5 ± 7.0 †‡ | 52.4 ± 7.6 † | 49.3 ± 9.4 † | 47.9 ± 10.7 † |
| EMG_BIS | C (n = 18) | 33.7 ± 6.1 | 32.5 ± 5.1 | 31.8 ± 4.2 | 31.9 ± 3.9 | 30.7 ± 3.9 |
|  | R (n = 19) | 32.6 ± 3.0 | 31.3 ± 3.2 †‡ | 28.9 ± 2.1*†‡ | 28.0 ± 1.5 *†‡ | 28.4 ± 1.7 † |
| PLE | C (n = 18) | 53.4 ± 8.4 | 53.9 ± 8.3 | 51.0 ± 8.2 | 50.8 ± 8.0 | 51.0 ± 7.9 |
|  | R (n = 19) | 57.1 ± 8.1 | 52.5 ± 6.3 | 48.4 ± 5.1 † | 47.7 ± 6.2 † | 48.2 ± 6.0 † |
| EMG_PLE | C (n = 18) | 23.8 ± 9.5 | 19.6 ± 6.2 | 19.5 ± 6.3 | 18.8 ± 6.5† | 18.9 ± 6.3 † |
|  | R (n = 19) | 23.5 ± 9.2 | 18.4 ± 4.0 | 18.8 ± 3.4 | 18.4 ± 3.8 | 19.1 ± 4.6 |
| PLE1 | C (n = 18) | 0.61 ± 0.04 | 0.63 ± 0.05 | 0.67 ± 0.08 †‡ | 0.70 ± 0.06 †‡ | 0.71 ± 0.06 † |
|  | R (n = 19) | 0.60 ± 0.05 | 0.61 ± 0.05 †‡ | 0.64 ± 0.06 †‡ | 0.66 ± 0.07† | 0.69 ± 0.07 †‡ |
| PLE2 | C (n = 18) | 0.58 ± 0.11 | 0.56 ± 0.13 | 0.54 ± 0.12 | 0.51 ± 0.11 | 0.51 ± 0.11 |
|  | R (n = 19) | 0.56 ± 0.14 | 0.55 ± 0.12 | 0.49 ± 0.08 ‡ | 0.51 ± 0.10 | 0.47 ± 0.08 |
| TOF count | C (n = 18) | 4 (0) | 4 (0) | 4 (0) | 4 (0) | 4 (0) |
|  | R (n = 19) | 4 (0) | 4 (0) | 4 (2)‡ | 2 (4) *† | 0 (4) *† |
| TOF ratio | C (n = 18) | 106.3 ± 28.6 | 107.1 ± 28.7 | 107.4 ± 29.9 | 106.6 ± 29.3 | 105.7 ± 28.7 |
|  | R (n = 19) | 102.9 ± 27.0 | 57.1 ± 44.8 *†‡ | 16.1 ± 30.1 *†‡ | 2.8 ± 8.7 *† | 0.0 ± 0.0 *† |

Values are mean ± SD or median (interquartile range). BIS; Bispectral index, PLE; Phase lag entropy, EMG; Electromyography, TOF; Train of four, TR0, TR1, TR2, TR3, and TR4: immediately before, 1, 2, 3, and 4 min after injection of rocuronium (group R) or same volume of normal saline as rocuronium (group C), respectively.

*: P < 0.05, compared with group C.

†: P < 0.05, compared with TR0.

‡: P < 0.05, compared with previous value.

**The changes of BIS and PLE after rocuronium in patients under sedation based on BIS > 60 at TR0.** Nine patients in group C and six in group R had a BIS value < 60 at TR0. Because the original objective was to evaluate the effects of NMB on the BIS and PLE under sedation based on a BIS of 60–80, we selected 22 patients (9 from group C and 13 from group R), who had a BIS > 60 at TR0, for comparison of values at TR0 and TR4, when the TOF ratio was 0 reflecting NMB in all patients. The demographic data of these patients are shown in S2 Table. The BIS and EMG_BIS decreased significantly at TR4 compared to TR0 in both groups. ΔBIS (TR0 –TR4) and ΔEMG_BIS (TR0 –TR4) did not differ significantly between the two groups (Table 5). PLE and EMG_PLE decreased significantly at TR4 compared to TR0 only in group R and ΔPLE (TR0 –TR4) in group R (12.2 ± 6.9) was significantly greater than that in group C (4.3 ± 6.0), while no significant differences were observed in ΔEMG_PLE (TR0 –TR4) between the two groups. PLE1 and PLE2 increased significantly and decreased at TR4, respectively, compared to TR0 in both groups, but ΔPLE1 (TR0 –TR4) and ΔPLE2 (TR0 –TR4) did not differ significantly between the two groups (Table 5).

**EEG spectral analyses from saved EEG on PLEMTM from TR0 to TR4 in patients with sedation.** A distinct difference was observed when we plotted the group-median spectrogram (Fig 4A) and the power spectra (Fig 4B) from the 4-min EEG data. Power in the alpha, delta, and slow-wave frequency bands was greater in group R than in group C. We plotted the power spectra for each group at TR0 (Fig 5A) and TR4 (Fig 5B) separately to determine the effects of NMB, and changes in the slow and delta frequency bands were observed at TR4 compared to TR0 in group R. Table 6 shows the changes in the power of each EEG wave after NMB in patients under sedation. The power of the alpha wave at TR4 increased in groups C and R compared to TR0; however, beta and gamma wave power decreased. The power of the slow and delta waves at TR4 increased compared to that at TR0 only in group R. The power of the slow, delta, alpha, and beta waves in group R was significantly greater than that in group C.

**Table 5. Changes in BIS and PLE of patients with sedation.**

|  | Group | TR0 | TR4 | TR0—TR4 (Δ) |
|---|---|---|---|---|
| BIS | C (n = 9) | 66.8 ± 5.3 | 51.8 ± 6.4 † | 15.0 ± 6.1 |
|  | R (n = 13) | 63.1 ± 2.4 | 50.1 ± 11.2 † | 13.0 ± 12.4 |
| EMG_BIS | C (n = 9) | 35.8 ± 7.3 | 30.8 ± 3.7 † | 5.0 ± 5.4 |
|  | R (n = 13) | 33.6 ± 3.0 | 28.5 ± 1.7 † | 5.2 ± 3.4 |
| PLE | C (n = 9) | 57.2 ± 5.5 | 52.9 ± 5.3 | 4.3 ± 6.0 |
|  | R (n = 13) | 60.6 ± 5.7 | 48.4 ± 6.1 † | 12.2 ± 6.9* |
| EMG_PLE | C (n = 9) | 21.0 ± 7.7 | 16.6 ± 3.1 | 4.4 ± 6.1 |
|  | R (n = 13) | 24.5 ± 10.1 | 18.3 ± 4.7 † | 6.2 ± 10.4 |
| PLE1 | C (n = 9) | 0.61 ± 0.05 | 0.71 ± 0.07 † | -0.10 ± 0.05 |
|  | R (n = 13) | 0.61 ± 0.06 | 0.67 ± 0.06 † | -0.07 ± 0.09 |
| PLE2 | C (n = 9) | 0.61 ± 0.09 | 0.52 ± 0.07 † | 0.09 ± 0.09 |
|  | R (n = 13) | 0.60 ± 0.14 | 0.46 ± 0.06 † | 0.14 ± 0.15 |
| TOF count | C (n = 9) | 4 (0) | 4 (0) |  |
|  | R (n = 13) | 4 (0) | 0 (4) *† |  |
| TOF ratio | C (n = 9) | 117.6 ± 10.2 | 118.7 ± 11.2 |  |
|  | R (n = 13) | 101.1 ± 32.7 * | 0 ± 0 *† |  |

Values are mean ± SD or median (interquartile range). BIS; Bispectral index, PLE; Phase lag entropy, EMG; Electromyography, TOF; Train of four, TR0 and TR4; immediately before and 4 min after injection of rocuronium (group R) or same volume of normal saline as rocuronium (group C), respectively.

*: P < 0.05, compared with group C.

†: P < 0.05, compared TR0.

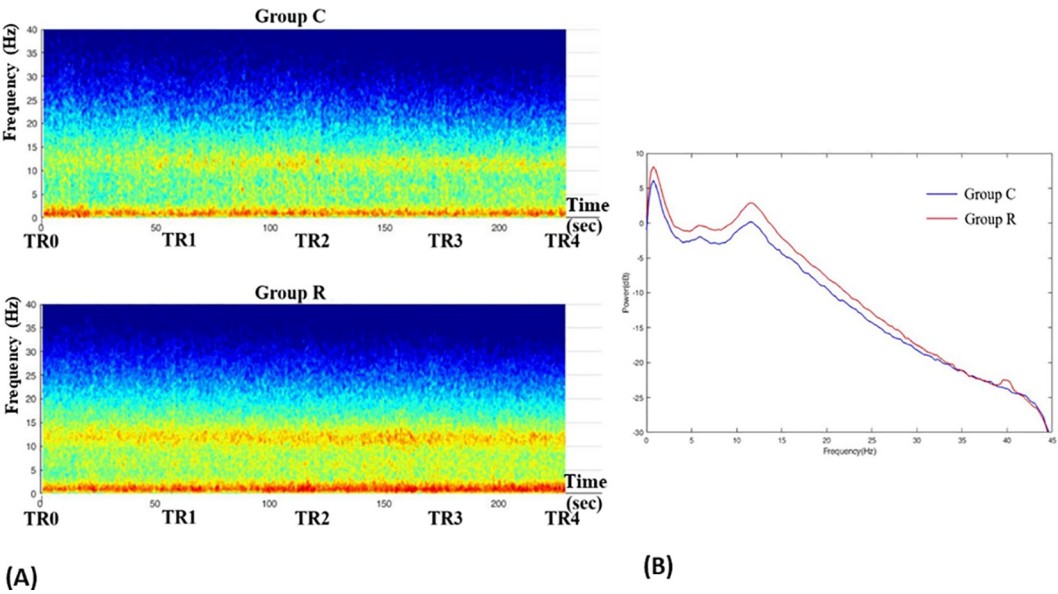

**Fig 4.** Group-median spectrogram (A) and power spectra (B) during TR0–TR4 comparing group C with group R. The group-median spectrograms (A) and power spectra (B) were plotted from the 4-min EEG data, and a distinct difference is evident. The power in the alpha, delta and slow-wave frequency bands was greater in group R than in group C.

### Study at the end of surgery

Table 7 shows the demographic data and clinical characteristics of groups D and B at the end of surgery. Fig 2B shows the time flow for the experimental procedure after the end of surgery.

**The changes of BIS and PLE for 4 minutes after administration of reversal agent of NMB under general anesthesia.** The propofol Ce values required to maintain the BIS at 40–55 in groups D and B were 3.1 ± 0.8 μg/ml and 2.8 ± 0.7 μg/ml, respectively, and no significant differences were detected between the two groups (P > 0.05). No significant changes in the

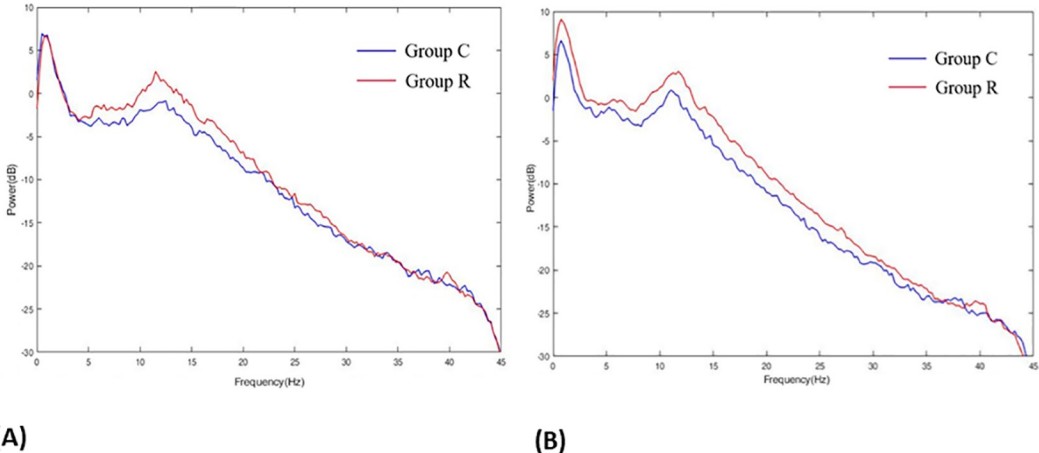

**Fig 5.** Power spectra at TR0 (A) and TR4 (B) comparing group C with group R. The power spectra for each group were plotted at TR0 (A) and TR4 (B) separately to determine the effect of NMB; changes in the slow and delta frequency bands were observed at TR4 compared to TR0 in group R.

**Table 6. Changes in power of each EEG waves after NMB in patients with sedation.**

|  | Group | TR0 | TR4 |
|---|---|---|---|
| Slow wave | C (n = 9) | 5.8 ± 2.4 | 5.2 ± 2.3 † |
|  | R (n = 13) | 5.4 ± 2.4 * | 8.1 ± 1.9 †* |
| Alpha wave | C (n = 9) | -2.4 ± 2.2 | -1.3 ± 2.3 † |
|  | R (n = 13) | -0.1 ± 2.0 * | 1.2 ± 2.1 †* |
| Beta wave | C (n = 9) | -9.7 ± 4.9 | -12.1 ± 5.3 † |
|  | R (n = 13) | -8.6 ± 5.1 * | -10.3 ± 5.4 †* |
| Delta wave | C (n = 9) | 0.04 ± 3.6 | 0.20 ± 2.8 |
|  | R (n = 13) | 0.36 ± 3.2 | 2.30 ± 3.4 †* |
| Gamma wave | C (n = 9) | -21.5 ± 3.7 | -24.2 ± 3.3 † |
|  | R (n = 13) | -21.4 ± 3.4 | -23.9 ± 3.7 †* |

Values are mean ± SD. TR0 and TR4; immediately before and 4 min after injection of rocuronium (group R) or same volume of normal saline as rocuronium (group C), respectively. Slow wave: 0.5–1 Hz, Alpha wave: 8–13 Hz, Beta wave: 13–30 Hz, Delta wave: 0.5–4 Hz, Gamma wave: > 30 Hz.

*: $P < 0.05$, compared with group C.

†: $P < 0.05$, compared with TR0.

BIS, PLE, EMG_PLE, PLE1, or PLE2 were observed at any time point compared to TRV0 (immediately before administration of the study drugs) in either group, only EMG_BIS increased significantly at TRV4 compared to TRV0 in group B (Table 8). Mean blood pressure and heart rate did not change significantly compared to TRV0, and no significant differences were observed between the two groups (P > 0.05) (S3 Table).

**The changes of BIS and PLE after complete recovery from NMB under general anesthesia.** At 4 min after administering sugammadex, all patients had a TOF ratio ≥ 100%. Therefore, we performed statistical analyses to evaluate the effects of complete recovery from NMB on the BIS and PLE during general anesthesia by comparing the values at TRV4 and TRV0. In groups D and B, no significant changes were observed in the BIS or PLE at TRV4 compared to TRV0, despite significant increases in EMG_BIS and EMG_PLE in group B (Table 9).

There was no unintended awareness or perioperative side effects related with study protocol in all patients.

**Table 7. Demographic and clinical data in the study at the end of surgery.**

|  | Group D | Group B |
|---|---|---|
|  | (n = 19) | (n = 18) |
| Age (years) | 46.6 ± 10.9 | 43.0 ± 12.9 |
| Gender (Male: Female) | 11: 8 | 10: 8 |
| Weight (kg) | 63.4 ± 10.3 | 63.5 ± 11.9 |
| Height (cm) | 166.1 ± 7.1 | 166.4 ± 11.4 |
| Surgical department (N, (%)) |  |  |
| OS | 10 (52.6) | 9 (50.0) |
| URO | 5 (26.3) | 3 (16.7) |
| GS | 3 (15.8) | 5 (27.8) |
| OBGY | 1 (5.3) | 1 (5.6) |

Values are mean ± SD or number (% of group). Group D: group who received same volume of normal saline as sugammadex, Group B: group who received sugammadex 2 mg/kg. There were no significant differences between the two groups.

**Table 8. Changes in BIS and PLE after administration of study drug at the end of surgery.**

| | Group | TRV0 | TRV1 | TRV2 | TRV3 | TRV4 |
|---|---|---|---|---|---|---|
| BIS | D (n = 19) | 45.9 ± 5.7 | 44.8 ± 5.6 | 43.8 ± 7.2 | 44.5 ± 8.4 | 43.9 ± 8.4 |
| | B (n = 18) | 45.9 ± 4.9 | 44.7 ± 5.3 | 43.5 ± 4.0 | 45.3 ± 5.7 | 45.7 ± 10.0 |
| EMG_BIS | D (n = 19) | 28.9 ± 3.5 | 28.6 ± 3.3 | 28.5 ± 3.4 | 28.8 ± 3.8 | 29.6 ± 5.6 |
| | B (n = 18) | 28.1 ± 2.2 | 28.3 ± 2.6 | 29.6 ± 3.5 | 30.4 ± 3.9 | 31.7 ± 5.4 † |
| PLE | D (n = 19) | 54.1 ± 12.4 | 54.3 ± 11.6 | 53.7 ± 11.6 | 54.2 ± 12.0 | 54.8 ± 10.8 |
| | B (n = 18) | 50.3 ± 6.3 | 49.4 ± 6.6 | 48.6 ± 8.4 | 49.8 ± 8.3 | 50.8 ± 8.8 |
| EMG_PLE | D (n = 19) | 20.5 ± 10.6 | 18.9 ± 10.4 | 19.5 ± 10.5 | 20.8 ± 10.1 | 21.6 ±10.8 |
| | B (n = 18) | 18.3 ± 3.8 | 21.1 ± 7.5 | 23.3 ± 10.7 * | 20.6 ± 4.1 | 22.8 ± 10.0 † |
| PLE1 | D (n = 19) | 0.75 ± 0.07 | 0.75 ± 0.07 | 0.76 ± 0.06 | 0.76 ± 0.05 | 0.78 ± 0.05 |
| | B (n = 18) | 0.77 ± 0.06 | 0.78 ± 0.05 | 0.79 ± 0.06 | 0.78 ± 0.06 | 0.79 ± 0.07 |
| PLE2 | D (n = 19) | 0.51 ± 0.12 | 0.53 ± 0.14 | 0.52 ± 0.12 | 0.52 ± 0.13 | 0.54 ± 0.12 |
| | B (n = 18) | 0.51 ± 0.08 | 0.50 ± 0.09 | 0.50 ± 0.11 | 0.54 ± 0.12 | 0.54 ± 0.12 |
| TOF count | D (n = 19) | 4 (0) | 4 (0) | 4 (0) | 4 (0) | 4 (0) |
| | B (n = 18) | 4 (0) | 4 (0) | 4 (0) | 4 (0) | 4 (0) |
| TOF ratio | D (n = 19) | 12.4 ± 19.3 | 14.9 ± 20.7 | 16.4 ± 20.6 | 16.3 ± 20.6 | 17.8 ± 20.4 † |
| | B (n = 18) | 10.0 ±7.3 | 52.2 ± 39.5 *†‡ | 91.1 ± 29.2 *†‡ | 107.5 ± 10.2 *† | 110.9 ± 10.1 *† |

Values are mean ± SD or median (interquartile range). BIS; Bispectral index, PLE; Phase lag entropy, EMG; Electromyography, TOF; Train of four, TRV0, TRV1, TRV2, TRV3, and TRV4: immediately before, 1, 2, 3, and 4 min after injection of sugammadex (group B) or same volume of normal saline as sugammadex (group D), respectively.

*: $P < 0.05$ compared with group D.

†: $P < 0.05$ compared with TRV0.

‡: $P < 0.05$ compared with previous value.

**Table 9. Changes in BIS and PLE of patients in general anesthesia.**

| | Group | TRV0 | TRV4 | TRV4—TRV0(Δ) |
|---|---|---|---|---|
| BIS | D (n = 19) | 45.9 ± 5.7 | 43.9 ± 8.4 | -2.1 ± 8.6 |
| | B (n = 18) | 45.9 ± 4.9 | 45.7 ± 10.0 | -0.17 ± 9.0 |
| EMG_BIS | D (n = 19) | 28.9 ± 3.5 | 29.6 ± 5.6 | 0.7 ± 4.8 |
| | B (n = 18) | 28.1 ± 2.2 | 31.7 ± 5.4 † | 3.7 ± 4.8 * |
| PLE | D (n = 19) | 54.1 ± 12.4 | 54.8 ± 10.8 | 0.7 ± 8.0 |
| | B (n = 18) | 50.3 ± 6.3 | 50.8 ± 8.8 | 0.5 ± 8.2 |
| EMG_PLE | D (n = 19) | 20.5 ± 10.6 | 21.6 ± 10.8 | 1.1 ± 6.9 |
| | B (n = 18) | 18.3 ± 3.8 | 22.8 ± 10.0 † | 4.5 ± 8.6 * |
| PLE1 | D (n = 19) | 0.75 ± 0.07 | 0.78 ± 0.05 † | 0.03 ± 0.06 |
| | B (n = 18) | 0.77 ± 0.06 | 0.79 ± 0.07 | 0.02 ± 0.03 |
| PLE2 | D (n = 19) | 0.51 ± 0.12 | 0.54 ± 0.12 | 0.03 ± 0.12 |
| | B (n = 18) | 0.51 ± 0.08 | 0.54 ± 0.12 | 0.03 ± 0.10 |
| TOF count | D (n = 19) | 4 (0) | 4 (0) | |
| | B (n = 18) | 4 (0) | 4 (0) | |
| TOF ratio | D (n = 19) | 12.4 ± 19.3 | 17.8 ± 20.4 † | |
| | B (n = 18) | 10.0 ± 7.3 | 110.9 ± 10.1 *† | |

Values are mean ± SD or median (interquartile range). BIS; Bispectral index, PLE; Phase lag entropy, EMG; Electromyography, TOF; Train of four, TRV0 and TRV4; before and 4 min after injection of sugammadex (group B) or same volume of normal saline as sugammadex (group D), respectively.

*: $P < 0.05$ compared with group D.

†: $P < 0.05$ compared with TR0.

## Discussion

The main findings of this study are as follows. The BIS, EMG_BIS, and EMG_PLE, but not PLE, decreased significantly in the eyes-closed condition during the awake state. The BIS and EMG_BIS decreased significantly at 4 min after administering rocuronium or the placebo in patients under sedation based on a BIS > 60; however, PLE and EMG_PLE did so only after NMB. Reversing the NMB did not affect the BIS or PLE under general anesthesia, despite significant increases in EMG_BIS and EMG_PLE at 4 min after administering sugammadex.

BIS is the most widely used tool to monitor the depth of sedation and anesthesia. It is based on power spectral and bispectral analyses of EEG acquired from a single channel [2]. Unlike the BIS, PLE, a recently developed consciousness-monitoring tool, measures the diversity of temporal patterns in the phase relationships between two signals and uses EEG data acquired from four channels (Fp1 and AF3; Fp2 and AF4) [6, 8]. PLE is useful as a hypnotic depth indicator during propofol sedation [8]. However, high-frequency signals, such as those from electrical devices, ECGs, and EMGs may contaminate the EEG signal and potentially increase the BIS [2, 3, 12, 13]. EMG is a known limitation of BIS monitoring because the BIS uses EEG signals up to 47 Hz, which includes EMG frequency bands 30–300 Hz and the BetaRatio calculation, a subparameter of BIS, includes the frequencies generated by the EMG [2, 3]. However, PLE has a different and unique algorithm and does not calculate a power value, which may lead to different effects of NMB on PLE than on the BIS. Based on previous results, in which NMB reduced the BIS during sedation under light general anesthesia, but not during general anesthesia, we performed this study at three different states of consciousness: awake, sedation, and general anesthesia.

### The effect of closing eyes on BIS and PLE during the awake state

The BIS, EMG_BIS, and EMG_PLE decreased significantly in the eyes-closed condition, but the PLE did not change significantly during the awake state, as expected. Although we did not measure EMG activity directly, the EMG activity of the frontalis muscle detected by the BIS and PLE monitors decreased during the eyes-closed condition, which led to a decrease in the BIS, but not PLE. This result is attributable to the difference in the algorithms of the two indices. As mentioned above, the BetaRatio used to calculate the BIS includes muscle activity frequencies [2, 3]; however, PLE does not adopt power values in its algorithm. PLE is calculated by the binary method based on the phase difference and EMG artefacts have less effect. The calculation of the PLE algorithm proceeds in three steps. Step 1: Obtaining the phase information of the measured signals (EEG and artefacts) from four channels (Fp1, Fp2, AF3 and AF4) [6]. Step 2: Binarization to the positive number, 1 and the negative number 0, and patterning by the number of the set parameter 'm.' Step 3: Calculation of entropy for pattern disposition. The main cause here is the second step. When a signal is generated by muscle activity and a common EMG signal is inputted to the two channels, the effects of the EMG are naturally eliminated during the binarizing of the phase difference (1, 0) (S2 Fig). That is, the EMG changes the value of the phase difference in the two signals, but if a common noise signal enters the two channels it is neglected when calculating and patterning the phase difference, so it does not greatly affect the phase direction of the positive or negative number. EEG signals are contaminated by eye-blink artefacts owing to the proximity of the eyes to the brain. Therefore, rejecting eye-blink artefacts from EEG signals has been a major challenge in the field of EEG signal processing [7]. When PLE detects blinking, it removes the large peak in the noise component and uses the remaining EEG component for the calculation. Thus, in the fully awake state the EMG of the frontalis muscle can affect the BIS, but not PLE.

## The changes of BIS and PLE after rocuronium in patients under sedation

**The changes of BIS and PLE for 3 min while propofol Ce at LOC was maintained.**  In 37 patients, propofol Ce at LOC was 3.5 ± 0.6 μg/ml. While the propofol Ce at LOC was maintained consistently for 3 min without rocuronium, the BIS, EMG_BIS, PLE, EMG_PLE, and PLE1, but not PLE2, continued to decrease significantly (Table 3). The power values of the alpha and beta waves became stronger and weaker, respectively, for 3 min according to the spectral analyses of the EEGs saved in PLEM™ (Fig 3). Propofol-induced unconsciousness induces a distinct spectral pattern: an increase in alpha, delta, and theta power and a decrease in beta and gamma power [14]. These results imply that the actual concentration of propofol in the brain, which cannot be measured, may increase despite a consistent propofol Ce being shown at the TCI pump, which could induce deeper sedation and decrease the BIS and PLE with a 3-min decrease in the EMG. Thus, unlike our initial plan, we failed to maintain the BIS at 60–80 for TR0 (immediately before the rocuronium injection) reflecting sedation in nine and six patients in groups C and R, respectively. Therefore, we selected 22 patients (nine from group C and 13 from group R), who had a BIS > 60 at TR0 to evaluate the effects of NMB on the BIS and PLE.

**The changes of BIS and PLE after NMB in patients under sedation based on BIS > 60 at TR0.**  In patients under sedation based on a BIS > 60, the BIS and EMG_BIS decreased significantly regardless of NMB by rocuronium; however, PLE and EMG_PLE did so only after NMB. Based on the results of previous BIS studies [2, 14, 15], we expected that NMB would decrease the BIS more than in the placebo group. We also expected that PLE would not change in groups C or R based on knowledge of the PLE algorithm [6]. However, our results differed from what we expected. Therefore, to investigate the cause, we performed EEG spectral analyses on the EEGs saved in the PLEM™ data from TR0 to TR4 in patients under sedation. The group-median spectrogram and power spectra from the 4-min EEG data showed that power was greater in group R than in group C in the alpha, delta, and slow-wave frequency bands (Fig 4). The power spectra created for each group at TR0 and TR4 to reveal the effects of NMB showed obvious changes in slow and delta frequency bands at TR4 compared to TR0 in group R (Fig 5). Taken together, the finding that NMB decreased PLE more (12.2 ± 6.9) than in group C (4.3 ± 6.0) (Table 5) implies that the results correspond to those from EEG spectral analyses. Unconsciousness induced by propofol is characterized by alpha, delta, and high-amplitude incoherent slow oscillations on EEGs [16, 17]. As shown in Table 6, NMB by rocuronium significantly increased the power of the slow and delta waves, but not in the placebo group, implying that NMB by rocuronium may deepen sedation by propofol, although the mechanism could not be elucidated. Some previous findings support this result that NMB decreases PLE. Forbes et al. [18] reported that pancuronium reduces halothane minimum alveolar concentrations by 25% and suggested both direct central effects and deafferentation theory as possible mechanisms. Deafferentation theory states that NMB abolishes muscle spindle afferent input to the reticular activating system, resulting in deafferentation of the cortex. This is supported by findings that cats paralyzed with an intravenous gallamine injection produce electrocortical synchronization, which is accompanied by the behavioral attributes of sleep, but not after an intracarotid injection, which implies no direct action on brain activity [19]. Schwartz et al. reported a pancuronium-induced dose-related increase in isoelectricity during EEG burst suppression in dogs anesthetised with isoflurane [20]. However, contradictory results have been reported, in which NMB did not alter the MAC of halothane in humans [21] and did not affect the depth of anesthesia measured using auditory evoked responses in either stimulated or unstimulated patients [22]. Another possible cause for a decrease in PLE after NMB compared to group C might have been the increased power of slow waves at TR4 in

group R, because the power value of the slow-wave is included in the PLE2 calculation, a sub-parameter of PLE.

The BIS and EMG_BIS decreased equally in groups R and C, indicating that NMB did not have an additive effect on the BIS or EMG_BIS. Several studies have shown that EMG activity falsely increases the BIS [13, 23] and that BIS decreases after administration of depolarizing and nondepolarizing NMBAs [3, 13, 15]. Bruhn et al. reported that the BIS decreased from 85 to 58 after administration of vecuronium, a nondepolarizing NMBA, at constant anesthetic drug concentrations [13]. Messner et al. reported that the lack of EMG activity caused by succinylcholine administration was accompanied by a steep decrease in the BIS from 96–97 to 33 or even to 9 in a volunteer with full consciousness [15]. The discrepancy in the results between their reports and this study may be attributed to a difference in the level of consciousness before administration of NMBA. Their cases were in a fully awake state or a lightly sedated state (BIS = 85) before NMB whereas the BIS before administration of NMBA was 60–80 in our patients. Because increasing frontalis muscle activity is mostly associated with light anesthesia [24] and none of these stimuli was applied in this study, the results of other studies might differ from ours if the research was carried out under a lightly sedated state or during application of tactile or painful stimuli, which can increase frontalis muscle activity. Therefore, further studies considering these aspects are needed.

Differences in software versions and the type of BIS monitor used may also be a possible explanation for discrepancies among studies. We used an A-3000 monitor with version 3.20 software, in which EMG and "near" suppression handling were improved [2]. However, Messner et al. used the A-1000 monitor with software version 3.31, in which EMG detection and removal were improved [2]. Indicators for EMG activity have been provided since the A-2000 monitor [15]. According to the manufacturer, the BIS detects the EMG_BIS at 70–110 Hz and displays four bars; one, two, three, and four bars represent the power at 30–38 dB, 39–47 dB, 47–55 dB, and >55 dB, respectively. The bar is displayed as empty in the EMG of ≤30 dB [25]. In our data, the lowest values of EMG_BIS and EMG_PLE at TR4 were 26 and 10, respectively, although these two EMG values cannot be compared. This implies no detection of a lower EMG by the BIS equipment, which leads to a masking additive effect on the BIS by the NMB-induced decrease in EMG_BIS. Therefore, the results of this study might have been different if BIS had been measured using upgraded software.

Although EEG spectral analyses cannot explain the depth of sedation and anesthesia completely, our finding that the alpha and beta wave power at TR4 significantly increased and decreased, respectively, compared to TR0 in the control group does not exclude the possibility of a change in the depth of sedation between TR0 and TR4 without NMB. That might be thought to be a cause of the decrease in EMG_BIS in the control group.

**The changes of BIS and PLE after complete recovery from NMB under general anesthesia.** Reversing the NMB did not affect the BIS or PLE under general anesthesia, despite significant increases in EMG_BIS and EMG_PLE at 4 min after administering sugammadex. Several reports are available on the effects of NMBAs on the BIS during anesthesia. Greif et al. [4] reported that the BIS level and EMG tone were unchanged after administering mivacurium during general anesthesia using propofol, neither by an EMG artefact nor by decreasing afferent neuronal input. Lee et al. showed that the BIS remained unchanged, regardless of NMB, in patients under general anesthesia (BIS 40) [26]. These results support our finding that there was no effect of EMG on either the BIS or PLE during general anesthesia.

Considering our results that BIS is affected by EMG activity, but not PLE, in awake or sedation states, PLE may be more reliable than BIS in clinical situations where there is a possibility of EMG contamination, such as facial surgery or nose surgery. However, future researches on this are needed. In addition, when monitoring BIS or PLE, it is important for clinicians to

judge the trend of BIS or PLE values in relation to clinical state of the patient along with EMG activity and SQI.

The limitations of this study must be discussed. First, the number of patients used to evaluate the effect of NMB on the BIS and PLE during sedation was small in both groups because we failed to maintain the BIS at 60–80 immediately before administering rocuronium. Therefore, further studies including a larger number of patients are needed. Second, we obtained EMG activity from the BIS and PLE equipment, not from an actual electromyograph. EMG activity based on the BIS or PLE equipment and the EEG gamma band can affect each other because both the EMG_BIS and EMG_PLE are calculated by mathematical formulas. Therefore, analyses using EMG activity measured using electromyography could change the results. Third, because the effect of NMBA on EEGs has not been clarified, we could not explain the mechanism of the EEG spectral changes by NMB. Forth, the possibility that manual ventilation vis face masks affected the measured values of BIS and PLE can be raised. However, that is very unlikely because we performed mask ventilation during sedation, taking care not to affect the forehead muscle activity as much as possible and SQI of the BIS and PLE were carefully monitored, and the BIS and PLE values when the SQI was $\leq$ 80 were not adopted when collecting data.

In conclusion, the effect of electromyograph activity and/or neuromuscular blockade on BIS or PLE depends on the level of consciousness. An effect of EMG was evident on the BIS, but not on PLE, during the awake state. During sedation, the BIS decreased with a decrease in EMG_BIS regardless of NMB by rocuronium, but NMB decreased PLE, although the degree of decrease in EMG_PLE after NMB was similar to that after injecting a placebo. No effect of the EMG was observed on either the BIS or PLE during anesthesia.

## Supporting information

**S1 Checklist. CONSORT checklist.**
(DOC)

**S1 Fig. PLE sensor and BIS sensor attached at the same time.** Upper BIS sensor; 1: reference electrode (FPz); 2: ground electrode (F3); 3: FT9; 4: measuring electromyography activity of the frontalis muscle (AF7). Lower PLE sensor; L1: Fp1, R1: Fp2, L2: AF3, R2: AF4, C: ground electrode, T: reference electrode.
(TIF)

**S2 Fig. PLE signal extraction process in EMG change.** Artefacts from two channels were eliminated during the phase difference and binarization calculations.
(TIF)

**S1 Table. Changes in vital signs of patients after administration of study drug.**
(DOCX)

**S2 Table. Demographic and clinical data in patients under sedation based on bispectral index $>$ 60 at before administration of study drug.**
(DOCX)

**S3 Table. Changes in vital signs after administration of study drug at the end of surgery.**
(DOCX)

**S1 File. Trial study protocol-English.**
(DOCX)

**S2 File. Trial study protocol-Korean.**
(DOC)

## Acknowledgments

We are grateful to Kyoung-Soo Kim (neuroscience researcher, InBody, Seoul, Republic of Korea) for his support with the spectral and PLE data analyses.

## Author Contributions

**Conceptualization:** Sohee Jin, Hee Jung Baik, Youn Jin Kim.

**Data curation:** Sohee Jin, Sooyoung Cho.

**Formal analysis:** Sohee Jin, Hee Jung Baik, Kyoung Ae Kong.

**Funding acquisition:** Hee Jung Baik.

**Investigation:** Sohee Jin.

**Methodology:** Sohee Jin, Hee Jung Baik, Rack Kyung Chung, Youn Jin Kim.

**Project administration:** Hee Jung Baik.

**Supervision:** Hee Jung Baik, Rack Kyung Chung.

**Validation:** Hee Jung Baik, Kyoung Ae Kong, Youn Jin Kim.

**Visualization:** Sohee Jin, Sooyoung Cho.

**Writing – original draft:** Sohee Jin, Hee Jung Baik.

**Writing – review & editing:** Sohee Jin, Hee Jung Baik, Sooyoung Cho, Rack Kyung Chung, Kyoung Ae Kong, Youn Jin Kim.

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
