## [Decision Letter · Decision Letter 0]

3 Jun 2021

PONE-D-21-12511

The influence of neuromuscular blockade on phase lag entropy and bispectral index: A randomized, controlled trial

PLOS ONE

Dear Dr. Baik:

Thank you for submitting your manuscript to PLOS ONE. After careful consideration, we feel that it has merit but does not fully meet PLOS ONE’s publication criteria as it currently stands. Therefore, we invite you to submit a revised version of the manuscript that addresses the points raised during the review process.

We look forward to receiving your revised manuscript.

Kind regards,

Academic Editor

PLOS ONE

2. Please note that PLOS does not permit references to 'data not shown.' Authors should provide the relevant data within the manuscript, the Supporting Information files, or in a public repository. If the data are not a core part of the research study being presented, we ask that authors remove any references to these data.

3. Please include the CONSORT flow diagram (blank version available at http://www.consort-statement.org/consort-statement/flow-diagram) as your Figure 1.

4. Thank you for submitting your clinical trial to PLOS ONE and for providing the name of the registry and the registration number. The information in the registry entry suggests that your trial was registered after patient recruitment began. PLOS ONE strongly encourages authors to register all trials before recruiting the first participant in a study.

1) your reasons for your delay in registering this study (after enrolment of participants started);

2) confirmation that all related trials are registered by stating: “The authors confirm that all ongoing and related trials for this drug/intervention are registered”.

"HJB: received grant from InBody Corporation, and the Industrial Strategic Technology Development Program (N10047988, 2013) funded by the Ministry of Industry and Trade.

A neuroscience researcher (Kyoung-Soo Kim, from InBody)performed the spectral and Phase lag entropy data analyses."

6.Thank you for stating the following in the Competing Interests section:

"I have read the journal's policy and the authors of this manuscript have the following competing interests:

This work was supported by InBody Corporation, and the Industrial Strategic Technology Development Program (N10047988, 2013) funded by the Ministry of Industry and Trade. This research was made possible by support from InBody who gifted the PLEM100 device and PLEM-ES100 electrode."

Reviewers' comments:

Reviewer's Responses to Questions

**Comments to the Author**

1. Is the manuscript technically sound, and do the data support the conclusions?

Reviewer #1: Partly

Reviewer #2: Yes

Reviewer #3: Yes

Reviewer #4: Partly

2. Has the statistical analysis been performed appropriately and rigorously? 

Reviewer #1: No

Reviewer #2: Yes

Reviewer #3: Yes

Reviewer #4: No

3. Have the authors made all data underlying the findings in their manuscript fully available?

Reviewer #1: Yes

Reviewer #2: No

Reviewer #3: Yes

Reviewer #4: Yes

4. Is the manuscript presented in an intelligible fashion and written in standard English?

Reviewer #1: Yes

Reviewer #2: Yes

Reviewer #3: Yes

Reviewer #4: Yes

5. Review Comments to the Author

Reviewer #1: This randomized, controlled study was evaluated the effects of NMB on PLE and the BIS.

I would thank the opportunity to review this manuscript. There are some comments that I think would improve the understanding of future readers.

In this study, the effect of NMB on PLE and BIS during general anesthesia (BIS 40-55) was evaluated by administering sugammadex or saline at a TOF count of 3 or 4. Strictly speaking, this is thought to be the impact of the recovery of NMB on PLE and BIS. For example, if the level of neuromuscular block was changed from TOF count 4 to TOF count 0 or less (deep block or intense block) while maintaining BIS 40-55, the results may be different.

Please describe the possibility that manual ventilation through face masks affected the measured values after injection of rocuronium or saline (TR0, TR1).

L82-86: Please provide a reference.

L159-L162: This sentence is a bit confusing. TOF ratios below 20% include the depth of neuromuscular blockade at various levels of NMB from intense block to recovery. Also, what is the basis for maintaining the TOF ratio below 20% during general anesthesia?

L197-L209: Did you conduct a normality test on continuous data? Some data shown in the tables (especially in table 6 and TOF count) are obviously not normally distributed. Providing median and IQR seems to be more correct instead of the means and standard deviations.

L159, L165 At the end of the operation, patients were received sugammadex or saline. In addition, At the end of surgery, patients were received reversal agents or saline. When is the exact timing of “At the end of surgery”?

P19L193-197: Please describe in more detail about the sample size calculation.

L380,381: What do you think is the cause of the decrease in EMG_BIS after administration of placebo?

Discussion

Please describe the clinical implications of this study based on results.

Thank you.

Reviewer #2: This is a very interesting RCT examining the impact of NMB on PLE and BIS according to sedation status.

They are some comments worth mentioning for the authors attention.

The primary and secondary objectives should be stated at the end of intro. Its no clear what the secondary objectives and also mention any safety outcomes.

1) The section Materials and Methods –could better structured. For example, make sections like e.g Statistical analysis clear (From 197), sample size calculations, randomisation etc...

2) “The BIS and PLE data were saved directly from the monitors to a USB flash drive, downloaded to a computer, and opened as an Excel file to extract the necessary data and confirm the handwritten collected data.” – This is concerning, patient data safety. Was the data encrypted etc. who was in possession and were standard operating processes followed for this? Also what was the handwritten data?

3) Line 111 Under material and methods, can the authors state what the groups are (C, R, D and B). They mention it line 150,161 but should be earlier. Then see next comment below.

4) Line 111, the authors have mentioned a two level or two step randomisation process. It will be of benefit if the authors could be more explicit. Sounds like different randomisations at the beginning and also at the end. Was this dependent on any outcome which determined randomisation into Group D or B? Or at the onset where people randomised to the respective groups. Could this be factorial?

5) Was there an analysis plan and was the person who did the analysis blind to treatment allocation? This needs to be stated whatever the case.

6) Table 2, It is not recommended to test baseline characteristics in RCT, by design you expect any differences between the groups to be random.

Reviewer #3: This study dealt with a very interesting topic, and the manuscript was well written based on the results.

However, it is difficult to understand the clinical implications of these findings. To help readers understand, I recommend to describe the clinical implications of the results of this study in discussion section.

Reviewer #4: Dear Authors,

#fundamental question

Since BIS and PLE have different algorithms for calculating the primary target value, simply evaluating the absolute value between the two devices is not suitable as a method of correctly comparing sedation levels.

#possible ethical issue

"During sedation, authors had been injected rocuronium to compare the changes in the parameters."

Was the sedation level deep enough at this time? Given that there is another expression for “During Anesthesia,” it may not be a completely deep state of sedation, which may be unethical for the subjects.

#lack of background and primary hypothesis

The effect of muscle relaxants on the BIS value is well known, and many studies have recently been published on the effect of the muscle relaxant on the entropy value. In addition to the known knowledge, it is not clearly described what the hypotheses of the authors to be explored are. It is necessary to state the core hypothesis more clearly. Inferences are possible in context, but why and what is measured and compared is not clearly explained.

#some concerns in the method section

- In the case of Group D (saline), mask ventilation was performed for 4 minutes. 1) Was it difficult to maintain the airway? 2) Have you considered the possibility of muscle artifacts caused by mask attachment?

- There is insufficient evidence for sample size calculation It did not reveal what the hypothesis was, and there was no explanation as to how large the effect size was set. References suggested as the basis for calculating the number of samples are not suitable because there is also no clear hypothesis in the paper and the effect size is not suggested.

Hypothesis testing in this study lacks statistical evidence due to inadequate sample counting, and the conclusions are also lacking in scientific evidence. (Critical Point)

All comparative results presented by the authors are not statistically supported, so it is difficult to draw clear conclusions.

#data acquisition issue

The two types of sensors used in the study (BIS, PLE) were originally developed on the premise of single attachment, so if they are attached to a competitive location at the same time, can they be attached exactly at the location recommended by the manufacturer? Is the error range according to the mounting location acceptable?

Thanks.

6. PLOS authors have the option to publish the peer review history of their article (what does this mean?). If published, this will include your full peer review and any attached files.

Reviewer #1: No

Reviewer #2: No

Reviewer #3: No

Reviewer #4: **Yes: **Sangseok, Lee

---

## [Author Response · Author response to Decision Letter 0]

19 Aug 2021

Dear Julia Robinson:

We would like to thank you and the reviewers of the PLOS ONE for taking the time to review our article. We have made some corrections and clarifications in the manuscript after going over the academic editor’s and the reviewers’ comments. The changes are summarized below:

Academic Editor’s comments

Response: We have done that. 

2. Please note that PLOS does not permit references to 'data not shown.' Authors should provide the relevant data within the manuscript, the Supporting Information files, or in a public repository. If the data are not a core part of the research study being presented, we ask that authors remove any references to these data.

Response: We revised those points. 

We added S1 Table, S2 Table, and S3 Table. 

3. Please include the CONSORT flow diagram (blank version available at http://www.consort-statement.org/consort-statement/flow-diagram) as your Figure 1.

Response: We have done that.

4. Thank you for submitting your clinical trial to PLOS ONE and for providing the name of the registry and the registration number. The information in the registry entry suggests that your trial was registered after patient recruitment began. PLOS ONE strongly encourages authors to register all trials before recruiting the first participant in a study.

1) your reasons for your delay in registering this study (after enrolment of participants started);

2) confirmation that all related trials are registered by stating: “The authors confirm that all ongoing and related trials for this drug/intervention are registered”.

Response: 

1) We mentioned the reason for the delay in registering this study in the ‘Materials and methods’ section as follows. “This trial was registered at the Clinical Trial Registry of Korea (http://cris.nih.go.kr, KCT0003750) after enrollment of participants started due to delay in the preparation of document in English.” (Page 7, lines 108-111)

2) It is not applicable. 

"HJB: received grant from InBody Corporation, and the Industrial Strategic Technology Development Program (N10047988, 2013) funded by the Ministry of Industry and Trade.

A neuroscience researcher (Kyoung-Soo Kim, from InBody)performed the spectral and Phase lag entropy data analyses."

Response: We mentioned the role of funders in our cover letter.

6.Thank you for stating the following in the Competing Interests section:

"I have read the journal's policy and the authors of this manuscript have the following competing interests:

This work was supported by InBody Corporation, and the Industrial Strategic Technology Development Program (N10047988, 2013) funded by the Ministry of Industry and Trade. This research was made possible by support from InBody who gifted the PLEM100 device and PLEM-ES100 electrode."

Response: We mentioned that statement in our cover letter. 

Reviewers' comments:

Reviewer's Responses to Questions

Comments to the Author

1. Is the manuscript technically sound, and do the data support the conclusions?

Reviewer #1: Partly

Reviewer #2: Yes

Reviewer #3: Yes

Reviewer #4: Partly

2. Has the statistical analysis been performed appropriately and rigorously?

Reviewer #1: No

Reviewer #2: Yes

Reviewer #3: Yes

Reviewer #4: No

Response: We performed additional statistical analyses according to reviewers’ comments and revised those points. 

3. Have the authors made all data underlying the findings in their manuscript fully available?

Reviewer #1: Yes

Reviewer #2: No

Reviewer #3: Yes

Reviewer #4: Yes

Response: We presented data not shown in the original manuscript as files in supporting information. (S1 Table, S2 Table, and S3 Table)

4. Is the manuscript presented in an intelligible fashion and written in standard English?

Reviewer #1: Yes

Reviewer #2: Yes

Reviewer #3: Yes

Reviewer #4: Yes

5. Review Comments to the Author

Reviewer #1: This randomized, controlled study was evaluated the effects of NMB on PLE and the BIS.

I would thank the opportunity to review this manuscript. There are some comments that I think would improve the understanding of future readers.

In this study, the effect of NMB on PLE and BIS during general anesthesia (BIS 40-55) was evaluated by administering sugammadex or saline at a TOF count of 3 or 4. Strictly speaking, this is thought to be the impact of the recovery of NMB on PLE and BIS. For example, if the level of neuromuscular block was changed from TOF count 4 to TOF count 0 or less (deep block or intense block) while maintaining BIS 40-55, the results may be different.

Please describe the possibility that manual ventilation through face masks affected the measured values after injection of rocuronium or saline (TR0, TR1).

Response: We performed mask ventilation during sedation, taking care not to affect the forehead muscle activity as much as possible and the signal quality indices (SQI) of the BIS and PLE were carefully monitored. The BIS and PLE values were considered inaccurate when the SQI was ≤ 80, and these values were not adopted when collecting data. These points were described in the discussion section. (Page 38, lines 572-577)

L82-86: Please provide a reference.

Response: We have done that. (Page 5, line 84, Page 6, line 85) )

L159-L162: This sentence is a bit confusing. TOF ratios below 20% include the depth of neuromuscular blockade at various levels of NMB from intense block to recovery. Also, what is the basis for maintaining the TOF ratio below 20% during general anesthesia?

Response: We did not maintain the TOF ratio below 20% during general anesthesia. We maintained the TOF count ≤ 1-2 during surgery with continuous infusion of rocuronium and stopped continuous infusion of rocuronium about 30 min before end of operation in order not to delay administration of reversal agents to the patients at the end of operation. In other word, we induced TOF count to be 3-4 at the end of operation. Therefore, at the end of surgery all patients showed TOF count 3-4, however, some patients showed TOF ratio < 20% due to individual variation. 

To avoid confusion, “TOF ratio < 20%” was changed to “TOF count ≥ 3 or a TOF ratio < 20%”. (Page 10, lines 169-170)

L197-L209: Did you conduct a normality test on continuous data? Some data shown in the tables (especially in table 6 and TOF count) are obviously not normally distributed. Providing median and IQR seems to be more correct instead of the means and standard deviations.

Response: We performed a normality test on continuous data and revised the descriptions for statistical analysis and Tables. (Page 13, lines 216-217 & Tables 4, 5, 8, and 9)

L159, L165 At the end of the operation, patients were received sugammadex or saline. In addition, At the end of surgery, patients were received reversal agents or saline. When is the exact timing of “At the end of surgery”?

Response: Thank you for finding the error. It has been corrected as follows. 

“We recorded all of the same measurements as induction before and after drug administration for 4 min at 1-min intervals (TRV0, TRV1, TRV2, TRV3, and TRV4) and thereafter, patients in groups D and B were given 0.4 mg glycopyrrolate and 10 mg pyridostigmine or the same volume of normal saline, respectively, and were awakened from anesthesia by stopping the TCI.” (Page 10, lines172-174)

P19L193-197: Please describe in more detail about the sample size calculation.

Response: We revised this part as follows. 

“Applying the effect size (1.04) calculated by the change of BIS after NMB (relaxant group: 12.9 ± 6.2, placebo group: 6.2 ± 6.7) of a previous study [11], the number of patients needed to provide a significance level (α error) of 0.05 and a statistical power (1 – β error) of 0.8 was 32.” (Page 13, lines 210-213)

L380,381: What do you think is the cause of the decrease in EMG_BIS after administration of placebo?

Response: As mentioned already in ‘Discussion section, although EEG spectral analyses cannot explain the depth of sedation and anesthesia completely, our finding of EEG spectral analyses does not exclude the possibility of a change in the depth of sedation between TR0 and TR4 without NMB. This increase in depth of sedation might be thought to be a cause of the decrease in EMG_BIS in the control group. We added this point to ‘Discussion’ section as follows. 

“Although EEG spectral analyses cannot explain the depth of sedation and anesthesia completely, our finding that the alpha and beta wave power at TR4 significantly increased and decreased, respectively, compared to TR0 in the control group does not exclude the possibility of a change in the depth of sedation between TR0 and TR4 without NMB. That might be thought to be a cause of the decrease in EMG_BIS in the control group.” (Page 36-37, lines 539-544)

Discussion

Please describe the clinical implications of this study based on results.

Response: We described clinical implications in the ‘Discussion’ section as follows. 

“Considering our results that BIS is affected by EMG activity, but not PLE, in awake or sedation states, PLE may be more reliable than BIS in clinical situations where there is a possibility of EMG contamination, such as facial surgery or nose surgery. However, future researches on this are needed. In addition, when monitoring BIS or PLE, it is important for clinicians to judge the trend of BIS or PLE values in relation to clinical state of the patient along with EMG activity and SQI.” (Page 37-38, lines 556-561)

Reviewer #2: This is a very interesting RCT examining the impact of NMB on PLE and BIS according to sedation status.

They are some comments worth mentioning for the authors attention.

The primary and secondary objectives should be stated at the end of intro. Its no clear what the secondary objectives and also mention any safety outcomes.

Response: We mentioned those objectives clearly as follows.

“The primary objective of this study was to evaluate the effects of NMB on PLE and BIS according to sedation status. The changes in the BIS and PLE after administering rocuronium, an NMBA, in the sedation state (BIS 60–80) during induction of anesthesia were compared to those of the placebo group, and the changes after administering sugammadex, an NMB reversal agent, during a general anesthetic state (BIS 40–55) were also evaluated. The secondary objectives of this study were to evaluate the effect of closing eyes on the BIS, PLE, EMG on BIS monitor (EMG_BIS) and EMG on PLE monitor (EMG_PLE) in the awake state and the effect of NMB on the EMG_BIS and EMG_PLE during sedation state and general anesthetic state.” (Page 6, lines 92-101) 

1) The section Materials and Methods –could better structured. For example, make sections like e.g Statistical analysis clear (From 197), sample size calculations, randomisation etc...

Response: We made new sections such as ‘Randomization’, ‘Interventions’, ‘Outcomes’, and ‘Statistical analyses’.

2) “The BIS and PLE data were saved directly from the monitors to a USB flash drive, downloaded to a computer, and opened as an Excel file to extract the necessary data and confirm the handwritten collected data.” – This is concerning, patient data safety. Was the data encrypted etc. who was in possession and were standard operating processes followed for this? Also what was the handwritten data?

Response: The subject identification information in the saved data to USB flash drive and the handwritten data was encoded to protect personal information of participants in this study. We mentioned this point in ‘Materials and Methods’ section as follows.

“The subject identification information in the saved data to USB flash drive and the handwritten data was encoded to protect personal information of participants in this study.” (Page 11, lines 185-187)

3) Line 111 Under material and methods, can the authors state what the groups are (C, R, D and B). They mention it line 150,161 but should be earlier. Then see next comment below.

Response: We modified this part as follows.

 “The patients were randomized into group R (rocuronium 0.6 mg/kg injected intravenously) or C (same volume of saline) during induction of anesthesia. Separately, they were also randomized into group B (sugammadex 2 mg/kg injected intravenously) or D (same volume of saline) at the end of the operation.” (Page7-8, lines 117-120)

4) Line 111, the authors have mentioned a two level or two step randomisation process. It will be of benefit if the authors could be more explicit. Sounds like different randomisations at the beginning and also at the end. Was this dependent on any outcome which determined randomisation into Group D or B? Or at the onset where people randomised to the respective groups. Could this be factorial?

Response: Yes, we performed different randomization processes at the induction of anesthesia and at the end of operation, independently. To help readers understand, we mentioned the sentence, “These two randomization processes were independent each other.” (Page 8, lines 120-121)

5) Was there an analysis plan and was the person who did the analysis blind to treatment allocation? This needs to be stated whatever the case.

Response: Yes, the person who did the statistical analysis was blind to group allocation. We have made this point clear. “The patients, anesthesiologists involved in the study, who collected the data, and who performed statistical analysis were blinded to the group allocations.” (Page 8, lines 124-126)

6) Table 2, It is not recommended to test baseline characteristics in RCT, by design you expect any differences between the groups to be random.

Response: We modified the related descriptions in the text as follows. “The demographic data and clinical characteristics during induction of anesthesia are shown in Table 2.” It is applied to Table 7. Therefore, we deleted the sentence, “No significant differences were observed between the two groups.” (Page 15, lines 249-250 & Page 24, lines 351-352)

Reviewer #3: This study dealt with a very interesting topic, and the manuscript was well written based on the results.

However, it is difficult to understand the clinical implications of these findings. To help readers understand, I recommend to describe the clinical implications of the results of this study in discussion section.

Response: We described clinical implications in the ‘Discussion’ section as follows. 

“Considering our results that BIS is affected by EMG activity, but not PLE, in awake or sedation states, PLE may be more reliable than BIS in clinical situations where there is a possibility of EMG contamination, such as facial surgery or nose surgery. However, future researches on this are needed. In addition, when monitoring BIS or PLE, it is important for clinicians to judge the trend of BIS or PLE values in relation to clinical state of the patient along with EMG activity and SQI.” (Page 37-38, lines 556-561)

Reviewer #4: Dear Authors,

#fundamental question

Since BIS and PLE have different algorithms for calculating the primary target value, simply evaluating the absolute value between the two devices is not suitable as a method of correctly comparing sedation levels.

Response: I agree with you about this point. We originally did not intend to compare absolute value between two devices. Therefore, we revised the statements about objective of this study as follows.

 “The primary objective of this study was to evaluate the effects of NMB on PLE and BIS according to sedation status. The changes in the BIS and PLE after administering rocuronium, an NMBA, in the sedation state (BIS 60–80) during induction of anesthesia were compared to those of the placebo group, and the changes after administering sugammadex, an NMB reversal agent, during a general anesthetic state (BIS 40–55) were also evaluated. The secondary objectives of this study were to evaluate the effect of closing eyes on the BIS, PLE, EMG on BIS monitor (EMG_BIS) and EMG on PLE monitor (EMG_PLE) in the awake state and the effect of NMB on the EMG_BIS and EMG_PLE during sedation state and general anesthetic state.” (Page 6, lines 92-101) 

#possible ethical issue

"During sedation, authors had been injected rocuronium to compare the changes in the parameters."

Was the sedation level deep enough at this time? Given that there is another expression for “During Anesthesia,” it may not be a completely deep state of sedation, which may be unethical for the subjects.

Response: While we maintained the effect-site concentration of propofol at the time of loss of consciousness for 3 minutes, all patients were in deep sedation state to such an extent that breathing was suppressed and facial mask ventilation was required. There are some reports on the relation of BIS and recall. Liu et al reported that no patients were able to recall the pictures shown during sedation with a corresponding BIS value of 80.8 ± 8.3 (mean ± SD) and suggested that to minimize the possibility of intraoperative recall during propofol-induced sedation or anesthesia, the BIS value should be maintained below 80 (Anesth Analg 1997;84:185-9). Glass et al reported that the BIS value at which there is a 95% probability of no recall was 77 (95% CI; 72-83) during target-controlled infusion of propofol (Anesthesiology 1997;86:836-47). In our study, BIS at TR0 (before administration of rocuronium or saline) was below 60 in 15 patients, 60-69 in 18 patients, and 70-73 in 4 patients. And especially, the range of BIS in our patients from TR2 to TR4 was 23 – 65. 

And different modalities of stimulation (e.g., painful surgical stimuli) can be associated with variable efficacy in creating memory traces. However, all our patients were not exposed to any external stimuli during the study period. Actually, in this study there was no unintended awareness in all patients. 

#lack of background and primary hypothesis

The effect of muscle relaxants on the BIS value is well known, and many studies have recently been published on the effect of the muscle relaxant on the entropy value. In addition to the known knowledge, it is not clearly described what the hypotheses of the authors to be explored are. It is necessary to state the core hypothesis more clearly. Inferences are possible in context, but why and what is measured and compared is not clearly explained.

Response: As you mentioned, it is well-known that conventional entropy measured using Datex-Ohmeda entropy module has been affected by NMB because EMG arousal is abolished by NMB. However, PLE used in this study is a newly developed device, which provides adequate information about functional connectivity between the prefrontal and frontal regions of the brain and uses totally different algorithm from conventional entropy for quantifying hypnotic depth. That is why we planned to do this study. We already mentioned these points in the ‘Introduction’. 

Based on your comments, the statement on the research hypothesis has been revised as follows.

“Therefore, we hypothesized that NMB affects BIS, but not PLE, during sedation state. And we also hypothesized that NMB does not affect BIS and PLE during a general anesthetic state.” (Page 6, lines 89-91)

#some concerns in the method section

- In the case of Group D (saline), mask ventilation was performed for 4 minutes. 1) Was it difficult to maintain the airway? 

Response: We had no difficulty at all in performing mask ventilation on the patient, and the airway was maintained well. There was no patient with a drop in oxygen saturation or a rise in end-tidal carbon dioxide partial pressure above the normal range.

2) Have you considered the possibility of muscle artifacts caused by mask attachment?

Response: That's a good point. We performed mask ventilation, taking care not to affect the forehead muscle activity as much as possible and in the meantime, the signal quality indices of the BIS and PLE were carefully monitored. The BIS and PLE values were considered inaccurate when the SQI was ≤ 80, and these values were not adopted when collecting data. These points were described in the ‘Discussion’ section. (Page 38-39, lines 572-577) In fact, during mask ventilation, there was no sudden change in the EMG or the case where the SQI fell below 80.

- There is insufficient evidence for sample size calculation It did not reveal what the hypothesis was, and there was no explanation as to how large the effect size was set. References suggested as the basis for calculating the number of samples are not suitable because there is also no clear hypothesis in the paper and the effect size is not suggested.

Hypothesis testing in this study lacks statistical evidence due to inadequate sample counting, and the conclusions are also lacking in scientific evidence. (Critical Point)

All comparative results presented by the authors are not statistically supported, so it is difficult to draw clear conclusions.

Response: We revised the descriptions for sample size calculation, hypothesis, primary and secondary objectives and outcomes of this study as follows.

“Therefore, we hypothesized that NMB affects BIS, but not PLE, during sedation state. And we also hypothesized that NMB does not affect BIS and PLE during a general anesthetic state.” (‘Introduction’, Page 6, lines 89-91)

“The primary objective of this study was to evaluate the effects of NMB on PLE and BIS according to sedation status. The changes in the BIS and PLE after administering rocuronium, an NMBA, in the sedation state (BIS 60–80) during induction of anesthesia were compared to those of the placebo group, and the changes after administering sugammadex, an NMB reversal agent, during a general anesthetic state (BIS 40–55) were also evaluated. The secondary objectives of this study were to evaluate the effect of closing eyes on the BIS, PLE, EMG on BIS monitor (EMG_BIS), and EMG on PLE monitor (EMG_PLE) in the awake state and the effect of NMB on the EMG_BIS and EMG_PLE during sedation state and general anesthetic state.” (‘Introduction, Page 6, lines 92-101)

“The primary outcomes were variables of BIS, PLE, differences of BIS and PLE between TR0 and TR4 {Δ(TR0 – TR4)}, and differences of BIS and PLE between TRV0 and TRV4 {Δ(TRV0 – TRV4)}. The secondary outcomes included EMG_BIS, EMG_PLE, PLE1, PLE2, hemodynamic parameters, and the power of each EEG waves.” (‘Materials and Methods’, Page 11, lines 178-182)

“Applying the effect size (1.04) calculated by the change of BIS after NMB (relaxant group: 12.9 ± 6.2, placebo group: 6.2 ± 6.7) of a previous study [11], the number of patients needed to provide a significance level (α error) of 0.05 and a statistical power (1 – β error) of 0.8 was 32. Forty patients were enrolled to allow for a dropout rate of up to 20%.” (‘Materials and Methods’, Page 13, lines 210-214)

#data acquisition issue

The two types of sensors used in the study (BIS, PLE) were originally developed on the premise of single attachment, so if they are attached to a competitive location at the same time, can they be attached exactly at the location recommended by the manufacturer? Is the error range according to the mounting location acceptable?

Response: As shown in S1 Fig, the two sensors could be attached without interfering with their respective attachment positions, and the two values were kept stable during the study period.

We hope the revised manuscript will better meet the requirements of your journal for publication. We thank the editor and the reviewers of the PLOS ONE once again for the constructive review of our paper.

---

## [Decision Letter · Decision Letter 1]

2 Sep 2021

The influence of neuromuscular blockade on phase lag entropy and bispectral index: A randomized, controlled trial

PONE-D-21-12511R1

Dear Dr. Baik,

We’re pleased to inform you that your manuscript has been judged scientifically suitable for publication and will be formally accepted for publication once it meets all outstanding technical requirements.

Kind regards,

Young-Kug Kim, M.D., Ph.D.

Academic Editor

PLOS ONE

Reviewers' comments:

Reviewer's Responses to Questions

**Comments to the Author**

1. If the authors have adequately addressed your comments raised in a previous round of review and you feel that this manuscript is now acceptable for publication, you may indicate that here to bypass the “Comments to the Author” section, enter your conflict of interest statement in the “Confidential to Editor” section, and submit your "Accept" recommendation.

Reviewer #1: All comments have been addressed

Reviewer #2: All comments have been addressed

Reviewer #3: All comments have been addressed

2. Is the manuscript technically sound, and do the data support the conclusions?

Reviewer #1: Yes

Reviewer #2: Yes

Reviewer #3: Yes

3. Has the statistical analysis been performed appropriately and rigorously? 

Reviewer #1: Yes

Reviewer #2: Yes

Reviewer #3: Yes

4. Have the authors made all data underlying the findings in their manuscript fully available?

Reviewer #1: Yes

Reviewer #2: Yes

Reviewer #3: Yes

5. Is the manuscript presented in an intelligible fashion and written in standard English?

Reviewer #1: Yes

Reviewer #2: Yes

Reviewer #3: Yes

6. Review Comments to the Author

Reviewer #1: I have confirmed that the paper has been revised according to the review opinions. Congratulations to the authors, this is a good and interesting research.

Thank you.

Reviewer #2: (No Response)

Reviewer #3: I did re-review your manuscript titled, “The influence of neuromuscular blockade on phase lag entropy and bispectral index: A randomized, controlled trial” . The authors submitted a well-revisioned manuscript according to my comments, and I believe this manuscript is available for publication in this journal.

7. PLOS authors have the option to publish the peer review history of their article (what does this mean?). If published, this will include your full peer review and any attached files.

Reviewer #1: No

Reviewer #2: No

Reviewer #3: **Yes: **SANG HUN KIM

---

## [Editor Report · Acceptance letter]

6 Sep 2021

PONE-D-21-12511R1 

The influence of neuromuscular blockade on phase lag entropy and bispectral index: A randomized, controlled trial 

Dear Dr. Baik:

I'm pleased to inform you that your manuscript has been deemed suitable for publication in PLOS ONE. Congratulations! Your manuscript is now with our production department. 

Kind regards, 

on behalf of

Prof. Young-Kug Kim 

Academic Editor

PLOS ONE